# Long-Term Decline in Harvester Termites in Madagascar following Multiple Barrier Treatments with Fipronil against Migratory Locust

**Ralf Peveling**

Department of Environmental Sciences—Geoecology, University of Basel, 4056 Basel, Switzerland;
ralf.peveling@unibas.ch

**Abstract:** Mounds of the harvester termite *Coarctotermes clepsydra* (Sjöstedt) (Isoptera: Termitidae, Nasutitermitinae) are a typical feature of savanna woodlands in Madagascar. With densities of up to 300 termitaria ha$^{-1}$, this species provides key ecosystem services and is an important food source for wildlife. Following large-scale aerial blanket and barrier treatments with the insecticide fipronil to control an outbreak of migratory locust in the late 1990s, evidence emerged that *C. clepsydra* and related food webs were adversely affected. However, neither the scale nor the duration of the effects were known. The present ex post study investigated the recovery of *C. clepsydra* populations subjected to multiple barrier treatments against hopper bands in 1998 and 1999 at estimated cumulative dose rates of 1.7–3.4 g fipronil ha$^{-1}$. At the time of the survey in 2007, both the density of occupied termitaria (30.2 versus 106.8 mounds ha$^{-1}$) and mound occupancy (24.3% versus 70.0%) were significantly lower in repeatedly sprayed so-called hotspots than in unsprayed areas. The overall adverse effect (mortality in sprayed areas corrected for mortality in unsprayed areas) was $E = 64.4\%$. The main outcome of this study is a strikingly low resilience of *C. clepsydra* populations, which did not recover in hotspots within eight years, with likely repercussions on food webs. This study shows that the environmental benefits of barrier treatments are forfeited if the same areas are treated repeatedly during the same campaign. Recommendations are given for the mitigation of these risks.

**Keywords:** insecticide disturbance; barrier treatment; spray history; non-target effects; *Coarctotermes clepsydra*; recovery; resilience





## 1. Introduction

Madagascar is a global biodiversity hotspot and has a high level of endemism in most taxonomic groups. Termite diversity, in contrast, is low [1], with only 56 species known in 2003, with the majority of them being wood feeders [2]. The grasslands and savanna woodlands in the semi-arid southwestern part of the island are home to just one mound-building species: the grass-feeding harvester termite *Coarctotermes clepsydra* (Sjöstedt) (Isoptera: Termitidae, Nasutitermitinae), whose conspicuous conical-shaped termitaria can reach densities of 100–300 mounds ha$^{-1}$ [2,3]. *Coarctotermes* is one of three grass-feeding genera in the Nasutitermitinae and the only one in Madagascar [4]. Given their high biomass and exposed lifestyle during nocturnal grass-harvesting bouts, harvester termites are an important food source for both invertebrates [5] and vertebrates [6]. In Madagascar, they are also commonly collected for chicken feed [7,8].

Harvester termites share their habitat with the Malagasy migratory locust, *Locusta migratoria capito* Saussure, and the red locust, *Nomadacris septemfasicata* (Serville) (Orthoptera: Acrididae), and are therefore at risk of exposure to locust insecticides during control operations. During the last two major locust invasions, which lasted from 1997–2000 and 2013–2016, aerial control operations extended over 42,000 and 23,000 km$^2$, respectively [9,10], relying mainly on synthetic chemical insecticides and, during the 1997–2000 campaign, on barrier treatments with fipronil.

Barrier treatments target marching hopper bands, which cannot fly. The technique is based on barriers of sprayed vegetation lying hundreds of meters apart. It is considered environmentally benign because large tracts of land remain untreated [11,12]. While traversing the area, feeding hopper bands eventually pass through treated vegetation and die. During the 1997–2000 campaign, fipronil, a relatively persistent phenylpyrazole locust insecticide and also a potent termiticide [13,14], was the sole control agent for barrier treatments used at an operational scale [15]. Overall dose rates (i.e., rates applied over the total target area) of single barrier treatments ranged from 0.75–1.50 g active ingredient (a.i.) ha$^{-1}$ at a 1000–500 m barrier spacing, respectively. In heavily locust-infested areas, treatments were applied repeatedly within the same season, which accordingly led to higher cumulative dose rates.

Earlier studies have shown detrimental effects on non-target fauna of both single barrier (1.0 g a.i. ha$^{-1}$) and blanket (3.2–4.0 g a.i. ha$^{-1}$) treatments with fipronil [16]. Harvester termites were found to be particularly vulnerable, with their decline leading to a decline in termite and/or ant-feeding reptiles and small mammals [3]. Field studies in Senegal and Australia corroborated the adverse effects of blanket treatments on termites over a range of dose rates (Senegal: 2.0–5.0 and 10.0–12.0 g a.i. ha$^{-1}$; Australia: 1.25 g a.i. ha$^{-1}$), though these studies did not investigate food chain effects [17–19]. In contrast, a subsequent study in Australia focusing on wood-eating termites, which owing to their more cryptic lifestyle are less exposed to insecticides, found no hazards associated with barrier treatments using fipronil at dosages per unit area ranging from 0.25–1.25 g a.i. ha$^{-1}$ [20].

While the fipronil-induced decline in harvester termites in Madagascar was unequivocal, eventually leading to cancellation of the registration in 2006 (Arrêté n°4196/06, 23 Mars 2006), its duration and spatial scale were unknown. Rapid assessments in southwestern Madagascar in 2005 provided circumstantial evidence of low densities of live colonies of *C. clepsydra* in former barrier treatment areas while abandoned and to varying degrees degraded mounds were abundant (author's observation). The present ex post study followed from this evidence. It tested the hypothesis that densities and occupancies of *C. clepsydra* termitaria and morphological features differ among unsprayed (no barrier treatments recorded) and sprayed areas subjected to multiple (≥2) barrier treatments from November 1998 to May 1999. The overall goal was to draw from the Madagascar experience and to inform risk assessments of fipronil and control agents with similar ecotoxicity worldwide.

## 2. Materials and Methods

This study was conducted in southwestern Madagascar. Based on spray protocols from aircraft equipped with differential global positioning systems, all barrier treatment areas (targets) were mapped on a monthly basis from November 1998 to May 1999 by the European Union (EU) Food Security Programme in Madagascar [15]. The main reason was to inform the campaign organization and to demonstrate progress in the control of the invasion. Original shape files of the Geographical Information System were lost after the campaign, but hard copies of the monthly spray maps and copies of the original spray protocols for 33 (out of more than 200) barrier treatments were retrieved by the author at the Delegation of the EU in Madagascar in 2005. The present study focused on an area covering 32,180 km$^2$ where intensive control operations had taken place (Figure 1). The area includes parts of the Sakaraha province (West) and the Haut Plateau de Horombé (East) and the national parks of Zombitse Vohibasia (West) and Isalo (East). Spray areas were, for the most part, adjacent to each other and sprayed only once, but a considerable proportion overlapped, resulting in multiple treatments (≥2) within the same campaign. The present study focused on these hotspots. The underlying assumption was that if full recovery of harvester termites was achieved in hotspot areas, the same would hold for the entire barrier treatment area.

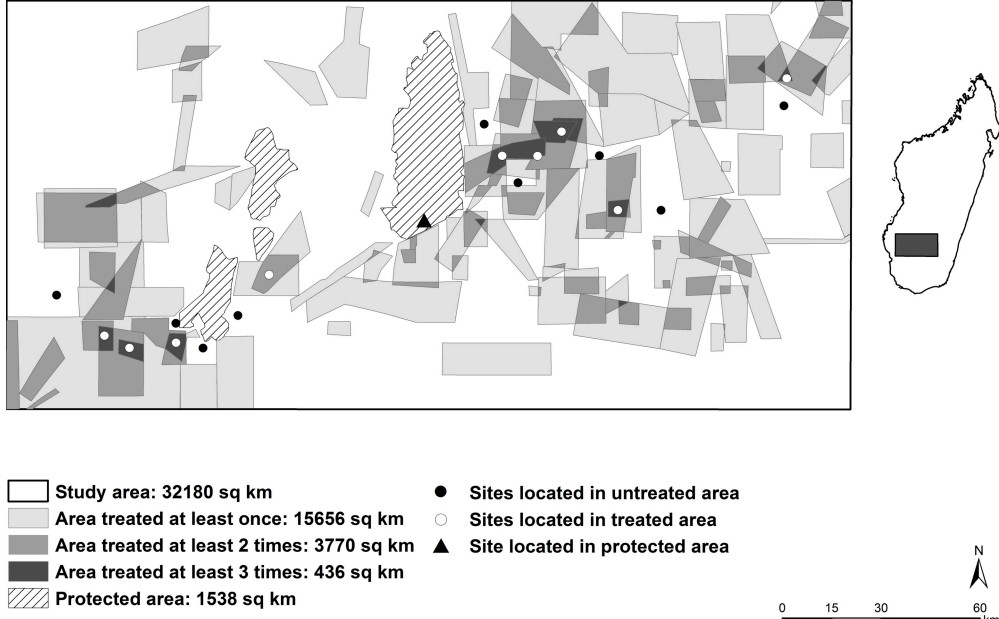

**Figure 1.** Study area and barrier treatment areas (targets) in southwestern Madagascar. Areas sprayed at least three times (dark grey) are included in areas sprayed at least twice (grey) and the latter are included in areas sprayed at least once (light grey). The design included 9 randomly selected pairs of sites—or blocks—located in untreated areas (black circles) and repeatedly treated hotspots (white circles), respectively. At each site, the densities, occupancies, and morphologies of termitaria were measured. At the single site located in Isalo national park (black triangle), only mound morphologies were measured.

According to the registered use of fipronil (Adonis® 7.5 UL) for barrier treatments in 1998, the maximum distance between barriers was 1000 m (Ministère d'Agriculture, Décision 166/98). No minimum distance was given at that time, but a decree issued by the Centre National Antiacridien in 2005 stipulated that the inter-barrier space should be 10 times as wide as the barrier (Note de service n°117-05/MAEP/CNA/Dir). According to available spray protocols ($N = 33$), the median barrier spacing was 875 m (range: 500–1000 m). At this distance, cumulative overall dose rates of fipronil correspond to ≈0.9, 1.7, 2.6, and 3.4 g a.i. ha$^{-1}$, respectively, at 1 single or 2, 3, and 4 overlapping barrier treatments. These values, however, are proxies because the actual combination of barrier spacing and the number of treatments was unknown for most targets.

A randomized paired-sample design was adopted, in which each site located in the center of a multiple barrier-treated hotspot was paired with a site located in an unsprayed area at a distance of 15–20 km (Figure 1). Sites were selected as follows: maps of barrier spray areas were scanned and imported into ArcGIS 9.4 (Esri Company, Redlands, CA, USA). They were georeferenced using the coordinates of targets for which spray protocols were available. Maps depicting monthly spraying operations were overlaid to identify overlapping areas of targets (Figure 1). Sampling sites were arbitrarily placed in the approximate center of hotspots. Paired unsprayed sites were arbitrarily placed in adjacent unsprayed areas at a minimum distance of 2–3 km from the boundary of the nearest sprayed area. For the purpose of the study presented in this paper, 9 pairs of sites—or blocks—were selected. The farthest distance between blocks was more than 200 km (Figure 1). Before starting the field work, the selected sites were verified to be located in typical locust and harvester termite savanna habitats by overlaying the barrier treatment map on the digital land use map of Madagascar (BD 500, Zone UTM 38 k, Institut Géographique et Hydrographique National de Madagascar, Antananarivo, Madagascar). The fire-resistant spear grass (*Heteropogon contortus*) is by far the most dominant grass species in these fire-prone savannas [21].

Each site was composed of 5 plots measuring 1000 m² each. The plots marked the center (original coordinates) and 4 corners of 1.4 × 1.4 km quadrats, with the corners located at a distance of 1 km at angles of 30°, 120°, 210°, and 300°, respectively, from the center point. Of the designated five plots per site, four were intended to be sampled depending on accessibility. Another site comprising six plots was investigated in Isalo national park. This was not planned before. Termitaria located there offered the opportunity to measure mound morphologies in presumably undisturbed conditions (no insecticide use, no harvesting of termites by humans).

The field work was conducted in March 2007. The designated plots were approached by vehicle as close as possible and further on foot, using the go-to function of the Global Positioning System. The median deviation from target waypoints was 7 m. A 17.84 m rope with wooden pegs fixed to both ends was used to delineate plots. One peg was driven into the ground to mark the center. The rope was pulled to its full length and the opposite peg circulated to outline the perimeter of a circle measuring 35.68 m in diameter and 1000 m² in surface area. All termitaria or remnants thereof lying within the circle were investigated. Even fully degraded or overgrown termitaria could be identified because grasses grew taller on the nutrient-enriched soil and subterranean mound chambers were still detectable. Mounds on the boundary were included if the center lay within the circle. The following parameters were recorded: minimum and maximum basal diameter, maximum height and number of domes, mound occupancy, and traces of human use such as regularly shaped marks from spades used to dig up termites. Neighboring termitaria with basal diameters >1 m apart were treated as different colonies. To verify occupancy, mounds were probed using picks. The search was extended to subterranean parts if no specimens emerged. Specimens of workers and soldiers were stored in 80% alcohol.

Data from the four plots per site (only three plots in one site) were pooled and extrapolated to one hectare. Mound density data were log-transformed and percent occupancy data (equivalent to survival rates) arcsine-transformed prior to two-factor analysis of variance (ANOVA) without replication, with treatment as the fixed and block as the random factor. This analysis is similar to a paired sample *t*-test.

The adverse effect *E* (mortality in repeatedly sprayed areas corrected for mortality in unsprayed areas) was calculated as a descriptive statistic according to Schneider-Orelli [22], where the percentage mortality per site was calculated as 100% minus the survival rate in this site. Effect calculations normally require pre-spray data from both sprayed and unsprayed sites. These were obviously not available as this study was conducted at 1 point in time 8 years post-spray. However, because mound structures persist for many years after the death of a colony, pre-spray densities (i.e., the status ante) of harvester termite colonies could be approximated.

The mound characters of occupied termitaria were analyzed using one-way ANOVA. Basal diameter was calculated as the mean of the minimum and maximum values. Data were log+1-transformed to achieve homogeneity of variances as needed. Analyses included pooled data from repeatedly sprayed, unsprayed, and control (national park) areas. Differences among means were tested using the Student–Newman–Keuls (SNK) test. A two-way contingency table was used to examine the association between treatment (repeatedly sprayed versus unsprayed) and termite-collecting efforts (traces of human use versus no traces). All tests were conducted using PASW Statistics 18 (SPSS Inc. Hong Kong, China).

## 3. Results

### 3.1. Density and Occupancy of Termitaria

The mean density of combined occupied and abandoned termitaria in repeatedly sprayed areas (93.5 ± 17.1 mounds ha$^{-1}$, mean ± SE) was significantly less than that recorded in unsprayed areas (156.6 ± 26.4 mounds ha$^{-1}$; $F_{1,8} = 9.1$, $p = 0.017$). Likewise, the mean density of occupied termitaria was significantly less than in unsprayed areas (30.2 ± 15.1 ha$^{-1}$ versus 106.8 ± 18.6 mounds ha$^{-1}$; $F_{1,8} = 17.9$, $p = 0.003$) (Figure 2a). With ≤3 inhabited termitaria ha$^{-1}$, 4 out of 9 barrier treatment sites were virtually void of live

harvester termite colonies though remnants of termitaria were frequently found. Mound occupancy was 24.3 ± 8.8% (mean ± SE) in repeatedly sprayed areas and 70.0 ± 4.7% in unsprayed areas ($F_{1,8}$ = 15.2, $p$ = 0.005) (Figure 2b). The resulting adverse effect was $E$ = 64.4 ± 12.7% (mean ± SE), meaning that at nearly 8 years post-spray, populations were suppressed by this percentage.

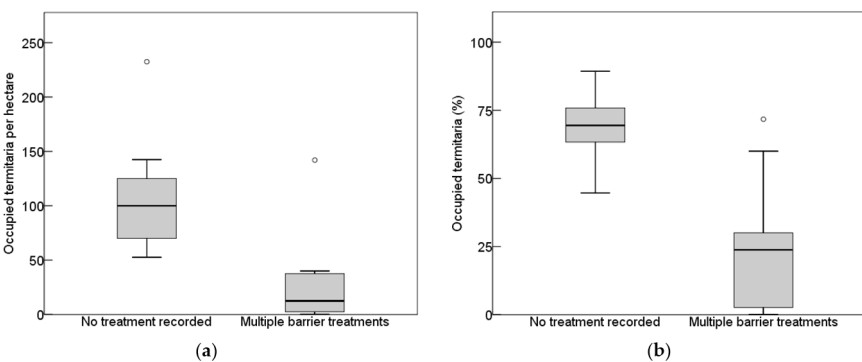

(a) (b)

**Figure 2.** Boxplots of the density (**a**) and percentage (**b**) of occupied termitaria in unsprayed (no treatment recorded) and repeatedly sprayed (multiple barrier treatments) areas (*N* = 9 in both treatments). Box plots show the median, lower/upper quartile, and smallest/largest observation and outliers (○). Differences among sprayed and unsprayed areas are statistically significant (see text).

*3.2. Mound Morphology*

The height of the occupied mounds was similar in sprayed (24.3 ± 1.8 cm; mean ± SE) and unsprayed (22.1 ± 0.9 cm) areas but differed significantly from the height recorded in undisturbed conditions in the Isalo national park (49.3 ± 4.5 cm; $F_{2,524}$ = 14.9, $p$ < 0.001, SNK test significant at $p$ < 0.05) (Figure 3a). Mound basal diameters were also similar among repeatedly sprayed (68.6 ± 3.9 cm) and unsprayed (70.7 ± 2.0 cm) areas but smaller than in undisturbed conditions (117.6 ± 8.1 cm; $F_{2,524}$ = 20.5, $p$ < 0.001, SNK test significant at $p$ < 0.05) (Figure 3b). The maximum number of domes per colony was 2. The mean number was 1.06 ± 0.04 in the national park and 1.07 in both unsprayed (SE = 0.01) and repeatedly sprayed (SE = 0.02) areas. These differences were not significant ($F_{2,455}$ = 0.02, $p$ = 0.980).

*3.3. Human Use of Harvester Termites*

Of all the occupied mounds in unsprayed areas (*N* = 377), 33.4% showed traces of anthropogenic activity. In most cases, the apex (where termites concentrate in the morning) had been cut off, resulting in a flat- or bowl-shaped top after repair. In repeatedly sprayed areas (*N* = 119), the percentage was only 17.6%, which was significantly less than expected (29.6%; *N* = 496, *d.f.* = 1, $\chi^2$ = 10.8, $p$ = 0.001).

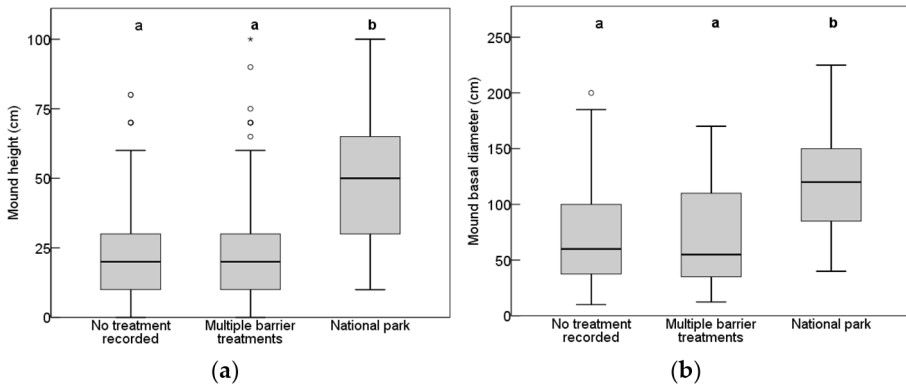

**Figure 3.** Boxplots of the height (**a**) and mean diameter (**b**) of occupied termitaria in unsprayed (*N* = 377; no treatment recorded) and repeatedly sprayed (*N* = 119; multiple barrier treatments) areas and in a national park (*N* = 31). Box plots show the median, lower/upper quartile, smallest/largest observation, outliers (○) and extreme cases (*). Means with different small letters are significantly different.

## 4. Discussion

The main outcome of the present research is a strikingly low resilience of *C. clepsydra* populations in hotspot areas. When exposed to 2–4 barrier treatments of fipronil within a single locust campaign, populations did not recover to pre-spray levels after 8 years. Large areas of intensive control were virtually cleared of *C. clepsydra*. This low resilience is likely due to the low dispersal capacity of termites. Dispersal distances generally do not exceed a few hundred meters [23] because alates are poor flyers [24]. Recolonizing vast deserted areas through repeated cycles of colony founding, growth, and dispersal would take a long time, depending on the density and spatial distribution of source colonies, and possibly also on the availability of abandoned but intact mounds for easier recolonization.

Another important outcome is that the environmental benefits of barrier over blanket treatments [11,12] are jeopardized if the same areas are sprayed repeatedly. The same applies if the barrier spacing is too narrow to provide truly unsprayed refugia from where recolonization can start. In an earlier study, even a 750 m barrier spacing resulted in a nearly 50% suppression in harvester termites after 2 years [3]. Considering that >30% of 33 barrier treatments for which spray protocols were available used a barrier spacing of only 500 m, adverse effects likely extended well beyond the 436 km$^2$ multiple barrier treatment hotspots studied in the present study. Extrapolating to the study area (Figure 1), 30% would correspond to about 3500 km$^2$. However, further research on *C. clepsydra* populations in single barrier treatment areas would have been required to test and quantify this assumption.

In any case, long-lasting reductions of harvester termites are expected to result in cascading effects across the trophic levels of affected ecosystems. This is because termites provide an array of ecological services, including decomposition of organic matter, nutrient cycling, enhancement of soil drainage, or bioturbation [25–28]. Moreover, termites influence the composition, structure, and spatial variation of vegetation [27,29,30] and the dynamics of ant communities [31], and mound-building termites play a keystone role by creating habitats for inquiline invertebrates [32]. The extent to which any of these processes was upset by the decline in *C. clepsydra* is not known. However, given its unique position as the only mound-building and grass-harvesting species of the savanna woodlands of Madagascar, perturbations were likely to be ecologically significant. At higher trophic levels, indirect effects of fipronil on wildlife are mediated mainly through loss in the quantity or quality of prey [33]. The above-mentioned study in Madagascar showed that the decline in harvester termites resulted in a reduced abundance of termite-feeding lizards and tenrecs [3]. A long-lasting decline in harvester termites over large areas would, therefore, affect the entire food web to which they belong.

In sub-Saharan Africa, termites and termite mounds are widely utilized by humans [8,34]. Examples include the consumption of protein-rich alates [35] or the use of mound material as fertilizer [36] or plaster for traditional houses [37]. In Madagascar and many other African countries, termites are frequently collected for chicken feed [7,8]. *C. clepsydra* colonies appear to recover well from recurrent damage to their mounds and loss in biomass, though this study showed that termitaria in both sprayed and unsprayed areas were smaller than in a protected area where the harvesting of termites is not allowed. An interesting finding was that inhabited termitaria in repeatedly sprayed areas were less frequently harvested than those in unsprayed areas. This suggests that it is less profitable to exploit a more widely dispersed resource because it requires more time and longer distances to find live colonies. It also shows that adverse effects on harvester termites directly translate into a negative impact on rural livelihoods.

An important methodological feature of this study was that inferences were made on pre-spray termitaria densities by combining occupied and abandoned termitaria. This was possible because remnants (e.g., subterranean mound chambers) or indications (e.g., circular grass patches different from surrounding grassland) of deserted mounds are detectable for many years. Moreover, the preponderance of deserted mounds in sprayed areas indicated that these had indeed been harvester termite habitats in the past.

However, the observed lower density of combined occupied and abandoned mound densities in repeatedly sprayed areas suggests that some fully weathered termitaria may have been undetected at the time of the field work. In this case, pre-spray densities and hence the adverse effect *E* would have been underestimated. There are only few studies on the persistence of abandoned mounds. A 10-year study on the growth of the Australian harvester termite *Drepanotermes perniger* (Froggatt) (Termitidae, Termitinae) showed that deserted mounds were not any more eroded than inhabited ones [38]. Lobry de Bruyn [39] hypothesized that mounds of *Drepanotermes tamminensis* (Hill) take at least 30 years to erode to ground level. However, the outer wall of *Drepanotermes* mounds is thicker and harder than that of *Coarctotermes* mounds (author's observation). Furthermore, mound erosion in southwestern Madagascar would be expected to be faster than in the Australian outback owing to much higher rainfall ($\approx$750 mm).

Conversely, the absence of epigeous mounds was considered an indication of the absence of harvester termites. This was reasonable because successful recolonization would have resulted in the development of visible mounds. Colonies of the ecologically similar harvester termite *Trinervitermes trinervoides* (Sjöstedt) (Nasutitermitinae) in South Africa begin building mounds 2–3 years after colony founding [40]. In view of this, significant visible recolonization would have been expected in sprayed areas after eight years.

Another question is whether wildfires may have had an effect on the outcome of this study. It is estimated that 450,000 ha of savanna are burnt in Madagascar each year [41]. However, there is no evidence that wildfires destroy colonies of harvester termites. On the contrary, grasslands dominated by fire-resistant spear grass are typical habitats of harvester termites. This is in line with studies in South Africa [42], Brazil [43], and Australia [44]. If wildfires supposedly have no effect on harvester termites, the reverse is not the case. With the densities of colonies and concomitant grass removal rates reduced, a larger proportion of the standing crop would be devoured by wildfires instead of being decomposed and recycled by termites.

## 5. Conclusions

Systemic insecticides, such as fipronil, have been linked to a variety of direct and indirect effects on vertebrate and invertebrate non-target organisms and ecosystems [33,45]. Insecticides that have adverse effects on populations of non-target invertebrates over prolonged periods are classified as high-risk products [11,46]. The results of the present and previous studies [3,16–18] leave no doubt that fipronil belongs to this category and that even barrier treatments can have long-lasting adverse effects on termites and related food webs. Therefore, rigorous and effective risk mitigation measures must be applied.

Apart from using less environmentally harmful insecticides for barrier treatment in the first place (e.g., benzoylurea insect growth regulators), risk mitigation measures should at least include the following: (i) mapping of historical and current locations of locust populations as a basis for improved monitoring, control, and impact assessment; (ii) monitoring of the magnitude and duration of adverse effects; (iii) restricting the use of fipronil or insecticides with similar ecotoxicity until full recovery of the affected non-target fauna has been evidenced; (iv) ensuring that the barrier spacing is wide enough to provide truly unsprayed refugia; and (v) imperatively avoiding multiple treatments.

**Funding:** This research received no external funding.

**Institutional Review Board Statement:** Not applicable.

**Informed Consent Statement:** Not applicable.

**Data Availability Statement:** The data presented in this study are available on request from the corresponding author.

**Acknowledgments:** I am grateful to Christian Rejela and Christale Robelle Razafindrahova for supporting the field work and to Clarah Andriamalala and Palvanijaz Pirnijasov for conducting GIS analyses. The Centre National Antiacridien (CNA) of Madagascar is acknowledged for providing logistical support and the EU Food Security Programme together with the Projet de Lutte Antiacridienne dans l'Aire Grégarigène for mapping barrier treatments during the 1998/99 campaign. I am also grateful to Harold van der Valk for his encouragement to undertake this study against all odds and without external funding. The late Colin C.D. Tingle and his team members in Madagascar and the UK are acknowledged for spearheading research on the environmental impact of fipronil and other locust control agents in Madagascar.

**Conflicts of Interest:** The author declares no conflict of interest.

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
