# Peer review of "Long-Term Decline in Harvester Termites in Madagascar following Multiple Barrier Treatments with Fipronil against Migratory Locust"

_agronomy, doi:10.3390/agronomy12020310_

Round 1

Reviewer 1 Report

This paper is well written and shows clearly that fipronil has a long-lasting effect on harvester termites. I have no hesitation in recommending that the ms should be published more or less as presented. A few minor comments:

  1. Line 118. "(fipronil)" should be added after "UL" to make it clear that Adonis is a trade name for fipronil,
  2.  Line 159. It is unclear how a termite mound can have a minimum and a maximum basal diameter. How was this distinction made? Also, when the results are presented in Lines 211-212 it is not stated which measurement is being referred to nor if it was, perhaps, the mean of the minimum and the maximum.
  3. Line 324. Delete "of time". "Period of time" is a tautology.
  4. Line 344 Replace "Data is" with "Data are".

Reviewer 2 Report

Review the manuscript agronomy-1435980

Peveling, R. Long-term decline of harvester termites in Madagascar following multiple barrier treatments with fipronil against Migratory Locust. Agronomy 2021, 11, x. https://doi.org/10.3390/xxxxx

The author is commended for intending to publish an older dataset which provides insight into the lack of recovery of termite populations in response to three or more applications of fipronil barrier treatments used in locust control. Surveys undertaken in both singly sprayed and multiple sprayed areas, with further replication within the national park would have greatly improved the design of this investigation, however, I can see that the findings are valuable, as there is very little literature which examines longer-term impacts of pesticides on non-target fauna. With this in mind, I would suggest minor revisions to the introduction to improve the arguments relating to the value of this work, as well as major revisions to the discussion. The discussion lacks structure and moves back and forth between topics of varied importance. Many sections needs improvement in focus and paragraph structure. I hope you find my detailed suggestions below helpful. There are also wording changes needed to conclusions and statements of the significance of the work, which are currently overstating the implications of the study. As the control site (national park) was under-sampled, and there are no ‘before’ surveys used in the analysis, it would be more acceptable to make conservative conclusions and instead of making strong recommendations, make suggestions for future work to improve knowledge in this area.

Detailed comments:

line 2: title change suggested "Lack of recovery of harvester termites..." as the sampling only occurred at one time point, and there is not a similar pre-spray survey or multiple post-spray surveys that would definitively measure a decline

Line 18: ‘lower in repeatedly sprayed’

Line 43: replace 'protected' with 'treated with locust control applications'

General comments on introduction: you should be clear that fipronil is no longer registered for use in Madagascar, and instead suggest the global value of your research is providing. It is also not clear until the methods that you only undertook comparisons between unsprayed and triple sprayed locations – there is not a gradient of spray treatments compared in this study.

Methods: figure 1 or methods should list the number of sites per treatment (untreated, treated three or more times) it seems there are 9 for each? This should be listed in all figures, as you do in the results. Could towns/villages be shown on the map to show the distribution of human activity levels? Also, the national park has only one site? It seems from methods, that there was only one site (6 plots, at one site?) within the national park. The area surveyed in the park is much less than the two treatments, therefore comparisons should be interpreted with caution, this should be noted in results/discussion. Finally, please describe any fire history, habitat or soil differences among sites that might also impact the size and occupancy of this termite species. Sites distributed across 200km are likely to have differences other than spray history and human activity levels.

Figure 3: thanks for sharing the sample sizes for the two treatments and national park, a comment about the power of the analysis being a bit low would be good to mention in the discussion.

line 252: are thicker

line 255: add a conclusion sentence that suggests that the above arguments support your assumption that your surveys are a valid measure of colony survival.

line 256: surveys for epigeous mounds were not mentioned in the methods - this should be added to the methods section

lines 265-268: As this is your main finding, i would move this paragraph to the start of the discussion. Leave the qualifiers to the second/third paragraphs.

Lines260-270: move this off-topic comment about the conservation status to the conclusion or combine with statements about ecological significance of the findings.

Lines 271-276: combine this with lines 265-268 to create a more complete first paragraph of the discussion.

Lines 277-284: The benefits of barrier over blanket treatments have not been well measured for termites using well designed before and after surveys, but maybe other taxa? These statements are lacking unequivocal support from multiple studies and your data. For this study, barrier spacing was not well recorded, as you state in methods, so i would suggest much more conservative language. This section might be moved to later in the discussion, and there is then an opportunity to suggest what future work should be undertaken to better address this question.

lines 285-295: great insight, i would move this paragraph towards the end of the discussion, after you discuss the ecological services (lines 301-309)

Lines 295-300: this is a main finding. move to earlier in the discussion - impacts on this species are likely to result in trophic cascades - use ecological literature to support this possibly. This might be a nice conclusion sentence or two to add to your first paragraph on the main finding of a lack of recovery.

lines 310-317: you did not mention the importance of fire in the intro or measure fire regimes at sites. i would cut this section - however, there is the possibly that factors such as fire, grazing, rainfall, soil type/quality and human land use differences impact termites, as hinted by the differences you found between the treatments and national park. I would replace this detailed discussion with a more general call to research the impacts of multiple factors on termite population ecology.

lines 324-328: I would qualify these strong statements to suggest that multiple (3 applications or more) are clearly detrimental. As you did not measure the areas impacted by only one or two applications, more surveys need to be undertaken to determine whether fipronil barrier treatments are a risk as single applications? Other research could be quoted here, and suggestions for future work made.

line 333: great point about the need to monitor when risky applications are used. I would quote reviews which has suggested the same approach, and mention that alternative locust control methods are available.

Lines 335-338: you raise an interesting point here - if fipronil treatments are no longer registered for use in Madagascar, can you comment on the value of you research for improvements to locust control globally?

Line 339-341: However, mapping could be improved by compulsory flight tracking of spray barriers - a key measure you were missing in your records? This is an odd final remark to make, would you instead want to choose to move this content earlier and combine with lines 329-334?
